

# A 3D deep learning approach to epicardial fat segmentation in non-contrast and post-contrast cardiac CT images

Thanongchai Siriapisith[1], Worapan Kusakunniran[2] and Peter Haddawy[2,3]

[1] Department of Radiology, Faculty of Medicine Siriraj Hospital, Mahidol University, Bangkok, Thailand
[2] Faculty of Information and Communication Technology, Mahidol University, Nakhon Pathom, Thailand
[3] Bremen Spatial Cognition Center, University of Bremen, Bremen, Germany

## ABSTRACT

Epicardial fat (ECF) is localized fat surrounding the heart muscle or myocardium and enclosed by the thin-layer pericardium membrane. Segmenting the ECF is one of the most difficult medical image segmentation tasks. Since the epicardial fat is infiltrated into the groove between cardiac chambers and is contiguous with cardiac muscle, segmentation requires location and voxel intensity. Recently, deep learning methods have been effectively used to solve medical image segmentation problems in several domains with state-of-the-art performance. This paper presents a novel approach to 3D segmentation of ECF by integrating attention gates and deep supervision into the 3D U-Net deep learning architecture. The proposed method shows significant improvement of the segmentation performance, when compared with standard 3D U-Net. The experiments show excellent performance on non-contrast CT datasets with average Dice scores of 90.06%. Transfer learning from a pre-trained model of a non-contrast CT to contrast-enhanced CT dataset was also performed. The segmentation accuracy on the contrast-enhanced CT dataset achieved a Dice score of 88.16%.

## INTRODUCTION

Epicardial fat (ECF) is localized fat surrounding the heart muscle and enclosed by and located inside the thin-layer pericardium membrane. The adipose tissue located outside the pericardium is called paracardial fat (*Bertaso et al., 2013*) and is contiguous with other mediastinal fat (Fig. 1). ECF is the source of pro-inflammatory mediators and promotes the development of atherosclerosis of coronary arteries. The clinical significance of the ECF volume lies in its relation to major adverse cardiovascular events. Thus, measuring its volume is important in diagnosis and prognosis of cardiac conditions. ECF volume can be measured in non-contrast CT images (NCCT) with coronary calcium scoring and in contrast-enhanced CT images (CECT) with coronary CT angiography (CCTA). However, accurate measurement of ECF is challenging. The ECF is separated from other mediastinal fat by the thin pericardium. The pericardium is often not fully visible in CT images,

Corresponding author
Thanongchai Siriapisith, thanongchai.sir@mahidol.edu

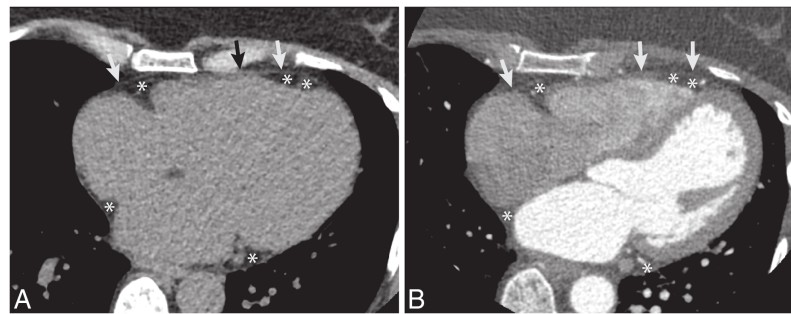

**Figure 1  The example CT dataset of epicardial fat.** (A) Non-contrast and (B) post-contrast CT images. The pericardium is a thin layer of membrane surrounding the heart (arrow). The epicardial fat is fat along the outer surface of heart and inside the pericardium (*).

which makes the detection of the boundaries of ECF difficult. ECF can also be infiltrated into grooves between cardiac chambers and is contiguous to the heart muscle. These technical challenges not only make accurate volume estimation difficult but make manual measurement a time-consuming process that is not practical in routine use. Therefore, computer-assisted tools are essential to reduce the processing time for ECF volume measurement.

Automated segmentation could potentially make ECF volume estimation more practical on a routine basis. Several approaches based on prior medical knowledge or non-deep learning techniques have been proposed for ECF segmentation, including genetic algorithms, region-of-interest selection with thresholding, and fuzzy c-means clustering (*Rodrigues et al., 2017*; *Militello et al., 2019*; *Zlokolica et al., 2017*). Deep learning techniques have been applied to a wide variety of medical image segmentation problems with great success (*Singh et al., 2020*; *Kim et al., 2019*; *Hesamian et al., 2019*). A recent article (*Renard et al., 2020*) demonstrates that deep learning algorithms outperform conventional methods for medical image segmentation in terms of accuracy. But most previous studies involved large solid organs or tumor segmentation (*Starke et al., 2020*; *Zhou, 2020*; *Turečková et al., 2020*). The segmentation of relatively small and complex structures with high inter-patient variability, such as ECF, has been far less successful. Recently, a few deep learning approaches to ECF segmentation have made progress on this problem (*He et al., 2020*; *Rodrigues et al., 2017*; *Commandeur et al., 2018*). In this paper, we build upon the previous work by presenting a novel deep learning model for 3D segmentation of ECF, We propose a solution of automatic segmentation of ECF volume using a deep learning based approach that is evaluated on both non-contrast and contrast-enhanced CT datasets. The NCCT dataset is from coronary calcium scoring and the CECT dataset is from contrast-enhanced coronary CT angiography (CE-CCTA) studies. The model is first learned from scratch on the NCCT dataset with coronary calcium scoring CT. To cover the entire heart, it is scanned in 64 slices with 2.5 mm thickness on each acquisition. Then, the model pre-trained on that NCCT dataset is transferred to the CECT dataset which is from CE-CCTA. The CE-CCTA study is performed in 256 slices with 0.625 mm thickness.

One of the key contributions of this paper is that we validate the performance of our newly developed 3D CNN-based approach on these difficult tasks. Since segmentation of ECF requires utilization of both voxel intensity and location information, we integrate two attention gate (AG) and deep supervision modules (DSV) into a standard 3D U-Net architecture. Our proposed model has better performance than the recent state-of-the-art approaches we evaluate because of the integration of AG and DSV modules. The AG module is used to focus on the target structures by suppressing irrelevant regions in the input image. The DSV module is used to increase the number of learned features by generating a secondary segmentation map combining the different resolution levels of network layers. The second main contribution is the use of transfer learning, taking a model pre-trained on NCCT data, and applying it to CECT data, using only a small amount of data for the re-training. This approach has benefits in clinical applications for both NCCT and CECT data for ECF segmentation. Furthermore, our proposed solution is 3D-based and does not require preprocessing and postprocessing steps, thus it can easily integrate into the clinical workflow of CT acquisition to rapidly generate ECF volume results for the physician in clinical practice.

## Related work

Conventional non-deep learning methods have been proposed for ECF segmentation. *Rodrigues et al. (2017)* proposed a genetic algorithm to recognize the pericardium contour on CT images. *Militello et al. (2019)* proposed a semi-automatic approach using manual region-of-interest selection followed by thresholding segmentation. *Zlokolica et al. (2017)* proposed local adaptive morphology and fuzzy c-means clustering. *Rodrigues et al. (2016)* proposed ECF segmentation in CECT images using the Weka library (an open-source collection of machine learning algorithms) with Random-Forest as the classifier. The experiment, performed on 20 patients, yielded a Dice score of 97.7%. However, these conventional methods required many preprocessing steps before entering the segmentation algorithm. The next evolution of ECF segmentations was performed with a deep learning approach. *Commandeur et al. (2018)* proposed ECF segmentation from non-contrast coronary artery calcium computed tomography using ConvNets. They reported the Dice score of 82.3%.

To improve the performance of medical image segmentation, several modifications of U-Net (*Ronneberger, Fischer & Brox, 2015*) have been proposed. The spatial attention gate has been proposed to focus on the spatial and detailed structure of the important region varying in shape and size (*Schlemper et al., 2019*). *Schlemper et al. (2019)* demonstrate the performance of the attention U-Net on real-time fetal detection on 2D images and pancreas detection on 3D CT images. *He et al. (2020)* proposed ECF segmentation from CE-CCTA using a modified 3D U-Net approach by adding attention gates (AG). AGs are commonly used in classification tasks (*Wu et al., 2020*; *Sharmin & Chakma, 2021*; *Fei et al., 2021*; *Kelvin et al., 2015*) and have been applied for various medical image problems such as image classification (*Kelvin et al., 2015*; *Zhao et al., 2017*), image segmentation (*Schlemper et al., 2019*; *Wu et al., 2020*; *Sharmin & Chakma, 2021*; *Fei et al., 2021*; *Kelvin et al., 2015*; *Zhao et al., 2017*), and image captioning (*Zhao et al.,*

*2017*). AG are used to focus on the relevant portion of the image by suppressing irrelevant regions (*Schlemper et al., 2019*). The integration of AG into the standard U-Net (*Hesamian et al., 2019*; *Turečková et al., 2020*; *He et al., 2020*; *Schlemper et al., 2019*; *Kearney et al., 2019*) or V-Net (*Turečková et al., 2020*; *Tureckova et al., 2019*) has been demonstrated to have benefits for region localization.

As mentioned above, the ECF has a complex structure. Some parts contain a thin layer adjacent to the cardiac muscle, which is similar to the microvasculature of the retinal vascular image visualized as small linear structures. In order to improve the performance of segmentation of small structures, several modules have been integrated into the main architecture of U-Net and V-Net, such as dense-layer and deep supervision modules (*Kearney et al., 2019*; *Tureckova et al., 2019*; *Liang et al., 2020*; *Wu, Zou & Zhan, 2019*; *Chen-Yu et al., 2015*). The dense-layer (*Liang et al., 2020*; *Wu, Zou & Zhan, 2019*) has been used to enhance the segmentation result instead of the traditional convolution in the U-Net model. Deep supervision (*Kearney et al., 2019*; *Tureckova et al., 2019*; *Chen-Yu et al., 2015*) was used to avoid local minimal traps during the training. The deep supervision helps to improve model convergence and increases the number of learned features (*Kearney et al., 2019*). *Kearney et al. (2019)* showed that addition of deep supervision added to the U-Net model can improve the performance of 3D segmentation in CT images of the prostate gland, rectum, and penile bulb.

While 2D and 3D deep learning approaches have been used for medical image segmentation, 3D approaches have typically shown better performance than the 2D approaches (*Starke et al., 2020*; *Zhou, 2020*; *Woo & Lee, 2021*). For example, *Zhou (2020)* demonstrated the better performance of 3D CNN approaches on multiple organs on 3D CT images, when compared to the 2D based method. *Starke et al. (2020)* also demonstrated that 3D CNN achieved better performance on segmentation of head and neck squamous cell carcinoma on CT images. *Woo & Lee (2021)* demonstrated that 3D U-Net provided better performance on brain tissue MRI images, compared with 2D U-Net, on a smaller training dataset. Therefore, in this paper we use a 3D CNN for segmenting epicardial fat in cardiac CT images.

## MATERIALS & METHODS

### CNN architecture

The model architecture is based on a 3D U-Net model composed of multiple levels of encoding and decoding paths. The initial number of features at the highest layers of the model is 32. The numbers of feature maps are doubled with each downsampling path. In addition to the original U-Net architecture, we added an attention gate connecting the encoding and decoding paths and deep supervision at the final step of the network. The model is created on a fully 3D structure at each network level. The final layer is an element-wise sum of feature maps of the two last decoding paths. The segmentation map of two classes (epicardial fat and background) is obtained using the output layer with threshold 0.5 to generate the binary classification of the epicardial fat. The architecture of the proposed network is shown in Fig. 2.

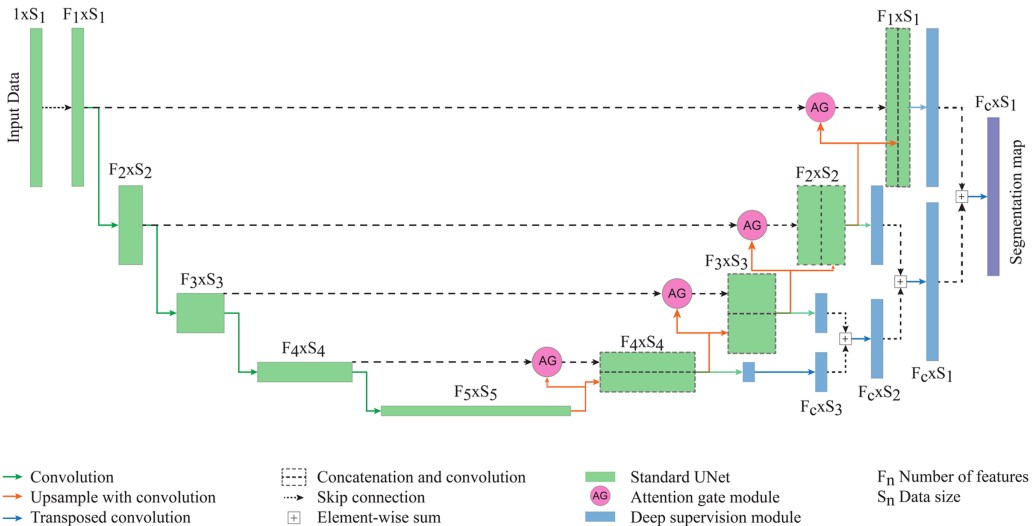

**Figure 2** **The proposed network for epicardial fat segmentation.** The network contains two main parts of the standard 3D U-Net integrated with the attention gate, and the deep supervision modules.

Starting with the standard 3D U-Net architecture, the attention gate module connects each layer of encoding and decoding paths. The gating signal (g) is chosen from the encoding path and the input features (x) are collected from the decoding path. To generate the attention map, g and x go through a $1 \times 1 \times 1$ convolution layer and element-wise sum, followed by rectified linear unit (ReLu) activation, a channel-wise $1 \times 1 \times 1$ convolutional layer, batch normalization and a sigmoid activation layer. The output of sigmoid activation is concatenated to the input x to get the output of the attention gate module (*He et al., 2020*; *Kearney et al., 2019*).

Deep supervision (*Turečková et al., 2020*; *Tureckova et al., 2019*) is the module at the final step of the network where it generates the multiple segmentation maps at different resolution levels, which are then combined together. The secondary segmentation maps are created from each level of decoding paths which are then transposed by $1 \times 1 \times 1$ convolution. All feature maps are combined by element-wise sum. The lower resolution map is upsampled by 3D transposed convolution to have the same size as the second-lower resolution. Two maps are combined with element-wise sum then upsampled and added to the next level of segmentation map, until reaching the highest resolution level.

## CT imaging data

This experimental study was approved and participant consent was waived by the institutional review board of Siriraj Hospital, Mahidol University (certificate of approval number Si 766/2020). The experimental datasets were acquired from 220 patients with non-contrast enhanced calcium scoring and 40 patients with CE-CCTA. The exclusion criteria were post open surgery of the chest wall. All CT acquisition was performed with the 256-slice multi-detector row CT scanner (Revolution CT; GE Medical Systems, Milwaukee, WI, USA). The original CT datasets of NCCT and CECT studies were 64 slices in 2.5 mm slice thickness and 256 slices in 0.625 mm slice thickness, respectively.

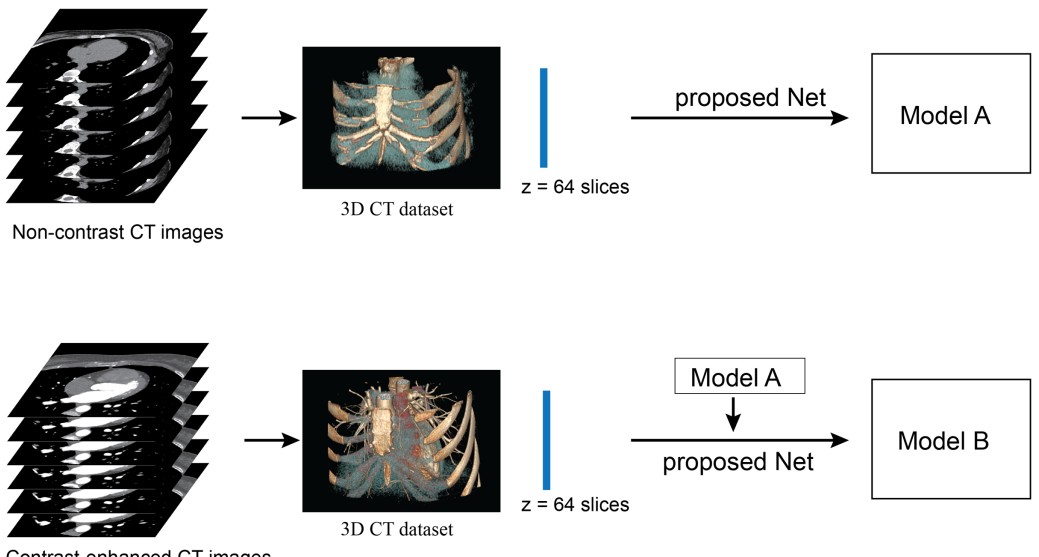

**Figure 3 Illustration of framework of the proposed method.** The upper row performs the network training from scratch with a non-contrast CT dataset. The lower row performs the network re-training on a contrast-enhanced CT dataset. No post-processing is required in this framework.

All DICOM images were incorporated into a single 3D CT volume file with preserved original pixel intensity. Due to limitation of GPU memory, the 256 slices of CE-CCTA were pre-processed with rescaling to 64 images in the volume dataset. The final 3D volume dataset in all experiments was $512 \times 512 \times 64$. The dataset was raw 12 bits grayscale in each voxel. The area of pericardial fat was defined by fat tissue attenuation inside the pericardium, ranging from −200 HU to −30 HU (*Rodrigues et al., 2016*; *Shahzad et al., 2013*; *Kazemi et al., 2020*). The ground-truth segmentation of ECF in all axial slices was performed using the 3D slicer software version 4.10.0 (*Fedorov et al., 2012*) by a cardiovascular radiologist with 18 years of experience. No additional feature map or augmentation was performed in the pre-processing step.

## Training framework

The experiments were implemented using the PyTorch (v1.8.0) deep learning library in Python (v3.6.9). The workflow for network training is illustrated in Fig. 3. The training and testing processes were performed on a CUDA-enabled GPU (Nvidia DGX-A100) with 40 GB RAM. The experiments were divided into three scenarios: model validity assessment, NCCT, and CECT experiments. The parameters were the same for all three experiments. The networks were trained with RMSprop optimizer and mean squared error loss. The training parameters of learning rate, weight decay, and momentum were le-3, le-8 and 0.9, respectively. The initial random seed was set to be 0. The illustration of the experimental framework is shown in Fig. 3.

The first experiment was the assessment of the model validity, for which we used five-fold cross validation. The total dataset consisted of 200 volume-sets (12,800 images), divided into five independent folds. Each fold contained 160 volume-sets (10,240 images)

for training and 40 volume-sets (2,560 images) for validation, without repeated validation data between folds. The other 20 volume-sets (1,280 images) were left for testing in the second and third experiments. Then the five-fold cross validation was performed on standard U-Net, AG-U-Net, DSV-U-Net, and the proposed method (AG-DSV-U-Net). For each fold of validation, the model with the best training accuracy after 150 epochs was selected for the validation.

The second experiment was to assess segmentation performance by training the network from scratch with the NCCT dataset. The volume matrix of each dataset was $512 \times 512 \times 64$ pixels. To compare the performance of segmentation, this experiment was performed with four model architectures: standard U-Net, AG-U-Net, DSV-U-Net, and the proposed method (AG-DSV-U-Net). The network was evaluated with the hold-out method, in which a total of 220 volume-sets (14,080 images) were split into 200 volume-sets (12,800 images) for training (the same used in the first experiment) and 20 volume-sets (1,280 images) for testing. The model output was collected at the maximum number of iterations at the 300th epoch, named model-A.

The third experiment was to assess segmentation performance on the CECT dataset and to evaluate the effectiveness of transferring the learning from NCCT to the CECT datasets. The pre-trained 3D model (model-A) was trained on the large calcium scoring NCCT datasets. The key to the success of the transfer learning with 3D U-Net is to fine-tune only the shallow layers (contracting path) (*Amiri, Brooks & Rivaz, 2020*) instead of the whole network. This contracting path represents the more low-level features in the network (*Amiri, Brooks & Rivaz, 2020*). The retraining dataset requires only a small amount of data-in our case only 20 volume-sets of CECT data. Note that these retraining cases are not from the same cases as used in the pre-trained model. To compare the performance of segmentation, this experiment was performed with four model architectures: standard U-Net, AG-U-Net, DSV-U-Net and proposed method (AG-DSV-U-Net). The network was evaluated with the hold-out method, in which the total 40 volume-sets (2,560 images) were split into 20 volume-sets (1,280 images) for training and 20 volume-sets (1,280 images) for testing. The output model is collected at the maximum number of iterations at the 300th epoch, named model-B.

## Performance evaluation

The performance of our proposed CNN segmentation is compared with the performance of the existing methods. The evaluation was quantitatively performed by comparison with the reference standard using the Dice similarity coefficient (DSC), Jaccard similarity coefficient (JSC), and Hausdorff distance (HD). An average HD value was calculated using the insight toolkit library of 3D slicer. Differences in the comparison coefficient among the four groups of experiments (standard U-Net, AG-U-Net, DSV-U-Net, and AG-DSV-U-Net) were assessed with a paired Student's t-test. P values < 0.05 indicated a statistically significant difference. Differences in the comparison between DSC of segmentation result and ECF volume were assessed with Pearson's correlation coefficient. The Pearson's values of <0.3 indicated poor correlation, 0.3 to 0.7 indicated moderate correlation, and >0.7 indicated good correlation. To assess the consistency of the

**Table 1 Patient characteristics of the CNN non-contrast and contrast-enhanced CT datasets.**

|  | Non-contrast CT | | Contrast-enhanced CT | |
| --- | --- | --- | --- | --- |
|  | Training dataset | Testing dataset | Training dataset | Testing |
| No. of records | 200 | 20 | 20 | 20 |
| Average age (years) | 61.41 ± 12.27 | 67.80 ± 11.66 | 65.85 ± 8.36 | 60.75 ± 10.31 |
| Average volume (ml) | 135.75 ± 60.09 | 127.59 ± 35.51 | 117.13 ± 69.29 | 121.43 ± 40.21 |
| Min volume (ml) | 6.39 | 71.86 | 47.34 | 66.03 |
| Max volume (ml) | 327.44 | 208.20 | 374.82 | 201.20 |

**Table 2 Results of five-fold cross-validation. Mean Dice score coefficient and standard deviation were used to assess model validity and repeatability on the non-contrast CT dataset.**

| Fold | U-Net | AG-U-Net | DSV-U-Net | AG-DSV-U-Net (Proposed method) |
| --- | --- | --- | --- | --- |
| 1 | 87.48 | 87.93 | 87.70 | 89.55 |
| 2 | 89.97 | 89.65 | 89.93 | 90.73 |
| 3 | 89.22 | 89.76 | 88.30 | 88.76 |
| 4 | 89.50 | 90.08 | 85.56 | 89.61 |
| 5 | 86.44 | 84.77 | 82.84 | 86.46 |
| Mean | 88.52 | 88.44 | 86.91 | 89.02 |

ground-truth, the inter-rater agreement was obtained by pixel-based correlation of DSC using two experienced cardiovascular radiologists with 18 and 14 years of experience. Ten volume sets (640 images) were randomly selected from NCCT dataset for the process of inter-rater correlation.

## RESULTS

The patient demographics are shown in Table 1. The training dataset consisting of 200 NCCT scan had an average age of 61.41 years and an average volume of 135.75 ml. The NCCT testing dataset had a similar distribution, with an average age of 67.80 years and an average volume of 127.59 ml. For the contrast-enhanced dataset, the average ages of the training and testing datasets were 65.85 and 60.75 years, respectively. The average volumes of ECF of the training and testing datasets were 117.13 and 121.43 ml, respectively. The inter-rater agreement of ECF ground-truth by two cardiovascular radiologists is about 91.73 ± 1.27%, which indicates excellent correlation.

Five-fold cross validation experiments on our NCCT dataset were used to evaluate the validity and repeatability performance of the proposed method. The dataset was split into training (80%) and validation (20%) for each fold. On each model architecture, the validation data exhibits good results across each fold. The proposed method also demonstrates the best average performance (DSC = 89.02), when compared with the other methods (at $p < 0.05$ when using validation data obtained in the five-fold cross-validation in the t-test), see Table 2.

**Table 3 Experimental results with standard 3D U-Net, AG-U-Net and the proposed method (AG-DSV-U-Net) on the holdout non-contrast CT dataset.**

| Non-contrast CT | U-Net | AG-U-Net | DSV-U-Net | AG-DSV-U-Net (Proposed method) |
|---|---|---|---|---|
| DSC | 84.87 ± 5.73 | 89.59 ± 4.45 | 89.70 ± 4.81 | 90.06 ± 4.60 |
| JSC | 74.12 ± 8.08 | 81.41 ± 6.77 | 81.64 ± 7.33 | 82.21 ± 6.91 |
| HD | 0.34 ± 0.18 | 0.27 ± 0.12 | 0.28 ± 0.14 | 0.25 ± 0.14 |

The experimental result on the hold-out NCCT test dataset is shown in Table 3. The proposed method demonstrates excellent results, achieving average DSC, JSC, HD values of 90.06 ± 4.60, 82.21 ± 6.91 and 0.25 ± 0.14, respectively. The baseline of the experiment is the standard 3D U-Net, which demonstrates good results with DSC, JSC and HD values of 84.87 ± 5.73, 74.12 ± 8.0.8 and 0.34 ± 0.18, respectively. The segmentation results of the modified U-Net models (AG-U-Net, DSV-U-Net, and the proposed method) demonstrate statistically significant improvement compared with the standard U-Net based on DSC ($p < 0.05$). The difference between segmentation results of AG-U-Net and DSV-U-Net is not statistically significant ($p > 0.05$). The DSC, JSC, HD values of AG-U-Net are 89.59 ± 4.45, 81.41 ± 6.77 and 0.27 ± 0.12, respectively. The proposed method statistically improved the segmentation result ($p < 0.05$) compared with AG-U-Net as well. While the proposed method is better than DSV-U-Net, the difference is not statistically significant ($p > 0.05$). Examples of segmentation results of the proposed method are shown in Fig. 4.

The experimental result of the proposed method on the CECT dataset is shown in Table 4. This transfer learning approach achieved average DSC, JSC and HD values of 88.16 ± 4.57, 79.10 ± 6.75 and 0.28 ± 0.20, respectively. The segmentation result of the proposed method demonstrates statistically significant improvement, when compared with the other methods based on DSC ($p < 0.0.5$). The segmentation results of the standard 3D U-Net and DSV-U-Net demonstrate good similar performance ($p > 0.05$). The segmentation results of the standard 3D U-Net and DSV-U-Net are statistically significantly better than those of AG-U-Net ($p < 0.05$). Examples of segmentation results of transfer learning with the proposed methods are shown in Fig. 5.

## DISCUSSION

Segmentation of ECF is a difficult image segmentation task because of the thin layer and complex structures at the outer surface of the heart. The ECF is also variable in distribution, depending on body habitus. In general, obese patients have larger amounts of ECF than do thin patients. Segmentation of ECF is more challenging than the segmentation of other cardiac structures.

Most CNN approaches work on 2D images whereas in clinical practice, 3D volume segmentation is used (*Milletari & Navab, 2016*). The 2D-based CNN approaches such as ResNet and VGG are not applicable for 3D datasets. The model architectures for 2D CNN and 3D CNN are different (*Starke et al., 2020*; *Zhou, 2020*; *Woo & Lee, 2021*). 3D CNN has an advantage over 2D-CNN by extracting both spectral and spatial features

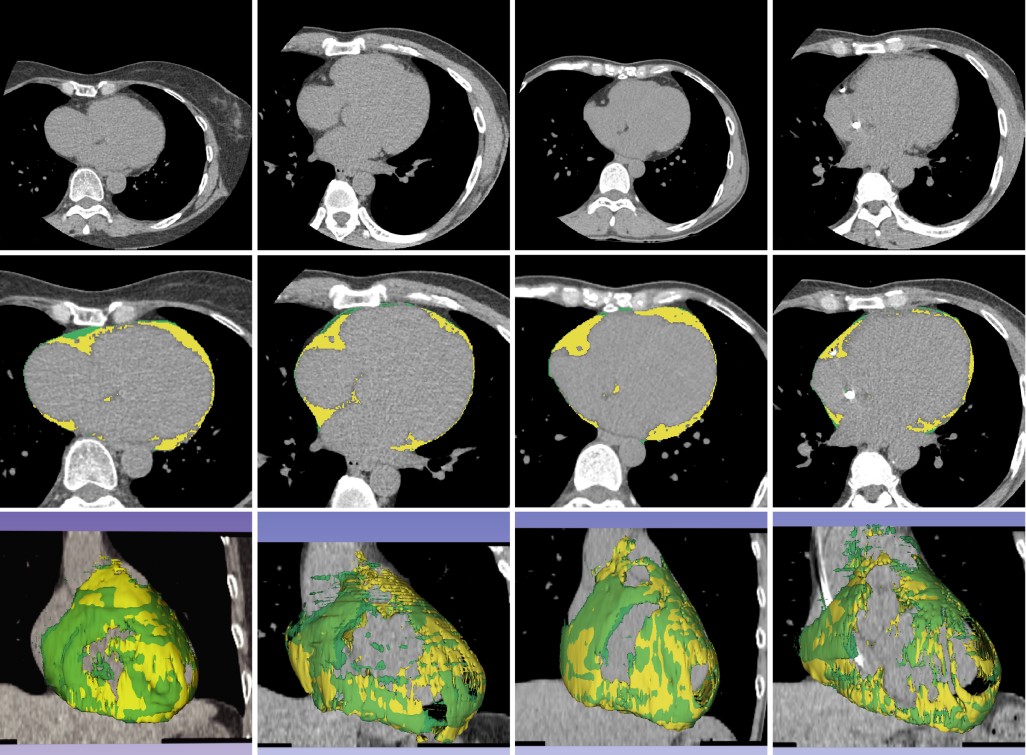

**Figure 4 Example of segmentation result of proposed method on non-contrast CT images.** The first, second, and third rows contain a source dataset in axial view, segmentation result in axial view, and segmentation result in 3D reconstruction, respectively. The yellow color represents the segmented ECF using the proposed method and green color represents the ground-truth. The DSC values for the volume sets from left to right are 90.90%, 88.63%, 95.31% and 87.86%, respectively. The fat volumes from left to right are 95.69, 116.95, 106.99 and 91.18 ml, respectively.

**Table 4 Transferred learning from pre-trained model to contrast-enhanced CT dataset.**

|  | U-Net | AG-U-Net | DSV-U-Net | AG-DSV-U-Net (Proposed method) |
|---|---|---|---|---|
| DSC | 85.58 ± 4.99 | 82.47 ± 4.33 | 85.07 ± 4.96 | 88.16 ± 4.57 |
| JSC | 75.11 ± 7.19 | 70.39 ± 6.03 | 74.32 ± 7.07 | 79.10 ± 6.75 |
| HD | 0.41 ± 0.36 | 0.34 ± 0.23 | 0.35 ± 0.30 | 0.28 ± 0.20 |

simultaneously, while 2D CNN can extract only spatial features from the input data (*Singh et al., 2020*). For this reason in general, the 3D CNN is more accurate than a 2D one (*Singh et al., 2020*; *Han et al., 2021*). 2.5D CNN has been developed to solve the memory consumption problem of 3D models (*Zhang et al., 2021*). 2.5D CNN has at least three approaches (*Zhang et al., 2021*; *Minnema et al., 2021*). The first is a combination of outputs of 2D CNNs in three orthogonal planes (axial, coronal, and sagittal) with majority voting. The second is to use 2D CNNs with three or five channels from adjacent three or five slices. Third is to apply 2D CNNs with randomly oriented 2D cross sections. In the final step, 2.5D segmentation requires an additional post-processing step to generate 3D output (*Han et al., 2021*). Although, the 3D CNN requires more resources and time for

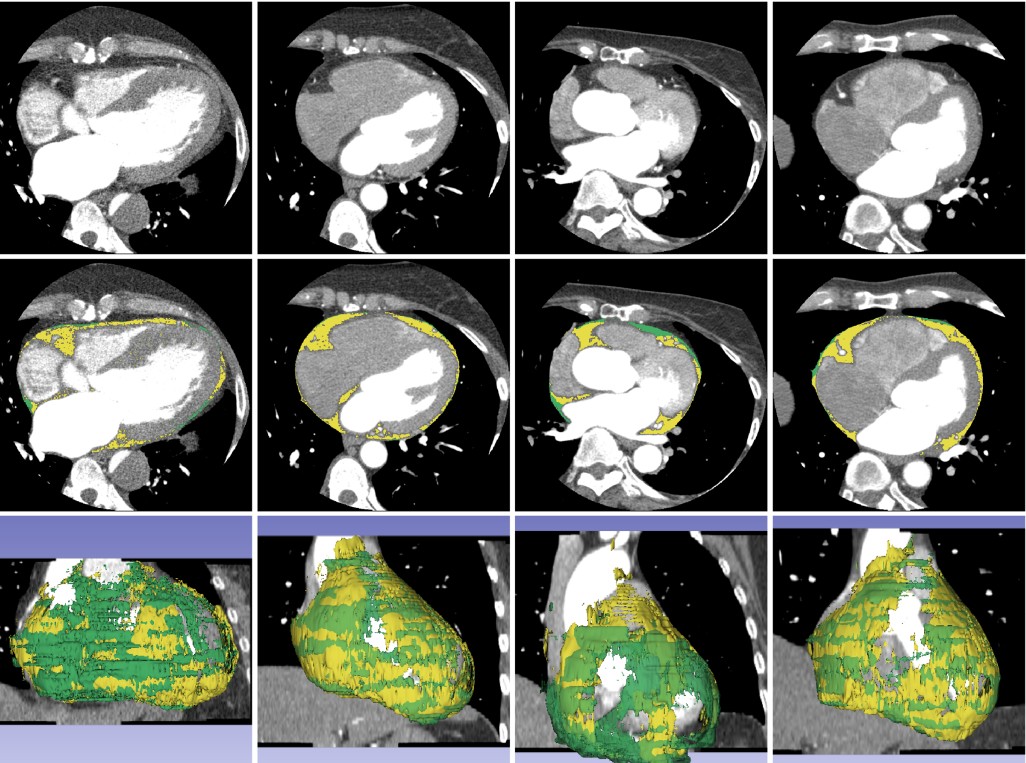

**Figure 5 Example of segmentation result of proposed method on contrast-enhanced CT images.** The first, second, and third rows contain a source dataset in axial view, segmentation results in axial view, and segmentation results in 3D reconstruction. The yellow color represents the segmented ECF and green color represents the ground-truth. The DSC values for the volume sets from left to right volume-sets are 80.23%, 93.38%, 72.26% and 92.72%, respectively. The fat volumes from left to right volume-sets are 201.20, 112.48, 92.28 and 112.98 ml, respectively. 

the model training, for the best performance, we use the 3D CNN in our implementation. The best performing methods for 3D volume segmentation of medical data are U-Net and V-Net. V-Net has more trainable parameters in its network architecture. Recent experimental comparisons of U-Net and V-Net on medical data have not shown statistically significant differences in performance (*He et al., 2020*; *Ghavami et al., 2019*). However, U-Net is less complex and easier to modify so that additional modules can be integrated into the standard U-Net in order to improve the performance.

Several state-of-the-art approaches for CNN-based segmentation of ECF have recently been proposed. *Commandeur et al. (2018)* proposed the first CNN-based method for ECF segmentation in a non-contrast 2D CT dataset using a multi-task convolutional neural network called ConvNets. They reported a Dice score of 82.3% for the segmentation result. *He et al. (2020)* proposed another CNN-based method on a 3D CECT dataset using AG integrated into 3D U-Net. The segmentation result was reported to have a DSC of 88.7% (*He et al., 2020*). We repeated the experiment by implementing the AG in 3D U-Net on our NCCT dataset by hold-out testing. The amount of training data (200 volume-sets) is more than the one used in the previous article (150 volume-sets) (*He et al., 2020*). Our four layer AG-U-Net method demonstrates

significantly improved performance with DSC of 89.59%, as compared with 3 layer AG-U-Net of 86.54% ($p < 0.05$). That should be due to more layers of our network. In our implementation, the AG-U-Net has more convolutional layers (four layers) and removes the sigmoid at the end of the network. However, the AG integration provides significantly better performance ($p < 0.05$) as compared with standard 3D U-Net (DSC of 84.87%). To the best of our knowledge, our experiment uses the largest volume size of any 3D CT dataset ($512 \times 512 \times 64$).

We introduce a novel approach to 3D segmentation of ECF by integrating both AG and DSV modules into all layers of the 3D U-Net deep learning architecture. The AGs are commonly used in natural image analysis and natural language processing (*Wang et al., 2017*; *Anderson et al., 2017*), which can generate attention-awareness features. The AG module is beneficial for organ localization, which can improve organ segmentation (*Schlemper et al., 2019*). During CNN training, AG is automatically learned to focus on the target without additional supervision (*Oktay et al., 2018*). The AG module can improve model accuracy by suppressing feature activation in irrelevant regions of an input image (*Schlemper et al., 2019*). The AG module is used to make connections between encoding and decoding paths on the standard U-Net. The DSV module is used to deal with the vanishing gradient problem in the deeper layers of a CNN (*Turečková et al., 2020*; *Chen-Yu et al., 2015*). The standard approach provides the supervision only at the output layer. But the DSV module propagates the supervision back to the earlier layer by generating a secondary segmentation map combining information from different resolution levels. The losses of this segmentation map are weighted and added to the final loss function and this can effectively increase the performance (*Kayalibay, Jensen & Smagt, 2017*). The DSV module is used by adding it into the decoding path of the 3D U-Net. The AG-DSV modules had been implemented in previous work (*Turečková et al., 2020*; *Tureckova et al., 2019*) for kidney (*Tureckova et al., 2019*) tumor segmentation (Kidney Tumor Segmentation Challenge 2019), as well as for liver (*Turečková et al., 2020*) and pancreas (*Turečková et al., 2020*) tumor segmentation (Medical Decathlon Challenge 2018).

The experiment demonstrated that our proposed method (AG-DSV-U-Net) achieves excellent performance with average and max DSC values of 90.06% and 95.32%, respectively. Our proposed method also shows a significant improvement of performance (90.06%), when compared with the previous state-of-the-art network (86.54%) on the same dataset ($p < 0.05$). The example of the results was shown in Fig. 4. While one might expect the segmentation performance to improve with fat volume of the dataset, unexpectedly, the statistical analysis demonstrates that there is poor correlation between segmentation performance and fat volume (Pearson's correlation 0.2).

The 3D volume size of our dataset is larger ($512 \times 512 \times 64$) compared with the previous work ($512 \times 512 \times 32$) (*He et al., 2020*). For comparative analysis on different numbers of slices and image resolution of datasets, the previous work demonstrates that a 40-slice of volume dataset achieves 1% higher DSC than 32-slice and

24-slice (*He et al., 2020*). However, the training time is also increased. We set up additional experiments to test the effect of different numbers of slices and image resolution on segmentation performance with our proposed model (AG-DSV-U-Net). The training, testing datasets, and hyperparameters are the same as defined in the NCCT experiment. The experiment of effect of the number of slices was performed by rescaling the slices with 64, 32 and 16 slices and using a fixed image resolution with $512 \times 512$ pixels. The segmentation results are DSC 90.06%, 81.76%, 78.93%, respectively. The experiment to determine the effect of different image resolution scales was performed by rescaling the resolution with $512 \times 512$, $256 \times 256$ and $128 \times 128$ pixels, with the number of slices fixed at 64. The segmentation results are DSC 90.06%, 86.19% and 83.73% respectively. The $512 \times 512$ image resolution and 64 slices still give the best performance, with significant improvement over lower resolution ($p < 0.05$). More slices and higher image resolution of the dataset let the network extract more spatial information that can help to improve segmentation accuracy. Furthermore, because the ECF is a thin layer along the heart contour, more spatial resolution will improve segmentation accuracy. To give the best performance, we chose 64 slices for our implementation which is a perfect fit with the original NCCT dataset, having 64 images in each dataset. In the CECT, the original CT dataset had 256 slices and needed to be rescaled to 64 slices. Due to the limitation of the proposed model and current GPU architecture, the voxel size of train and test datasets cannot be extended beyond 64 slices. The other limitation of this study is the size of the dataset: 220 volume-sets for NCCT experiment and 40 volume-sets for CECT experiment.

In clinical practice, the cardiac CT scan can be performed in NCCT or CECT or using both methods. For this reason, the ECF can also be either segmentation from NCCT or CECT dataset. To the best of our knowledge, ours is the first implementation of ECF segmentation on NCCT and CECT datasets. In our experiment, we started to train with the NCCT dataset (200 volume-sets). We used the concept of transfer learning to re-train with a similar dataset by taking a small amount of the dataset (Fig. 3). We re-trained the pre-trained NCCT model with a small amount of CECT data (20 volume-sets). We test the model with additional testing of 20 volume-sets. The experimental result shows good performance with a DSC value of 88.16%. The performance is also significantly better than that obtained with the standard U-Net and AG-U-Net (Table 4). Training from scratch is more generally applicable but requires a large amount of training samples. Both NCCT and CECT datasets are similar in appearance but differ in color. The training from scratch approach of both datasets required a large amount of data samples. The transfer learning method is more practical because it uses smaller amounts of training data and still yields good performance. Additionally, our proposed re-trained model demonstrates good performance as compared with the previous training from scratch (88.7%) (*He et al., 2020*).

Future studies could include investigations into more data diversity from multiple CT vendors, larger patient variation, and testing the model across different healthcare centers. Further investigation of clinical correlation between CNN segmentation of ECF volume and occurrence of cardiovascular disease is another interesting research questions.

## CONCLUSIONS

We have introduced a CNN-based approach for ECF segmentation using integration of AG and DSV modules into the standard 3D U-Net. ECF segmentation is one of the most difficult medical image segmentation tasks. We trained the NCCT dataset from scratch and also re-trained on a CECT dataset from the pre-trained NCCT model. We successfully improved the performance of ECF segmentation on both NCCT and CECT datasets when compared with the previous state-of-the-art methods. It is expected that this proposed method has the potential to improve the performance in other difficult segmentation tasks. The concept of training and retraining models can be also applied to other medical image segmentation problems.

## ACKNOWLEDGEMENTS

Thank you to Dr.Jitladda Wasinrat who is the second cardiovascular radiologist for analysis of inter-rater agreement.

### Funding

This work was supported by a grant from the Mahidol University Office of International Relations to Haddawy in support of the Mahidol-Bremen Medical Informatics Research Unit. The funders had no role in study design, data collection and analysis, decision to publish, or preparation of the manuscript.

### Grant Disclosures

The following grant information was disclosed by the authors:
Mahidol University.

### Competing Interests

The authors declare that they have no competing interests.

### Author Contributions

- Thanongchai Siriapisith conceived and designed the experiments, performed the experiments, analyzed the data, performed the computation work, prepared figures and/or tables, authored or reviewed drafts of the paper, and approved the final draft.
- Worapan Kusakunniran conceived and designed the experiments, authored or reviewed drafts of the paper, and approved the final draft.
- Peter Haddawy conceived and designed the experiments, authored or reviewed drafts of the paper, and approved the final draft.

### Ethics

The following information was supplied relating to ethical approvals (*i.e.*, approving body and any reference numbers):

This experimental study was approved by the institutional review board of Siriraj Hospital, Mahidol University (certificate of approval number: Si 766/2020).

## Data Availability

The source code and data is available in the Supplemental Files. The raw data is available at figshare: Siriapisith, Thanongchai; Kusakunniran, Worapan; Haddawy, Peter (2021): A 3D deep learning approach to epicardial fat segmentation in non-contrast and post-contrast cardiac CT images. figshare. Dataset. https://doi.org/10.6084/m9.figshare.17039813.v2.

## Supplemental Information

Supplemental information for this article can be found online at http://dx.doi.org/10.7717/peerj-cs.806#supplemental-information.

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
