# Peer review of "A 3D deep learning approach to epicardial fat segmentation in non-contrast and post-contrast cardiac CT images"

_PeerJ Computer Science, doi:10.7717/peerj-cs.806_

## Round 0.1 · original submission · Major Revisions

This seems a promising submission, but the reviewers have some concerns regarding the experimental design that must be addressed. Please address all the comments of the reviewers in the revised version.

Reviewer 1 ·

Basic reporting

This research suggests a U-net based CNN segmentation for epicardial fat, which is very challenging task.
Overall, the submitted manuscript seems to have reasonably described the related works and the necessity of the research.
However, I have a few concerns about the methods, the evaluation , and the discussion .
First, the materials and methods part does not seem to be written clearly. The author had better describe more clearly the contribution of their research. And, I suggest the authors that the materials and Methods part should have some sub sections for readability.
Second, I am not sure why the author presented the experimental result of 5-fold cross validation. Are the other experimental results not the cross validation results?
Then, the author had better report how to divide the training set and the test set.
Currently, it does not seem to be clear whether all the experimental results were cross validation based results or not.
Third, in the discussion part, the author had better analyze and infer why such promising results were produced. For example, attention gates and deep supervision module were used in your research. The author had better describe why those gate and module made the performance improved precisely.

Experimental design

As I mentioned above, the author presented the experimental result of 5-fold cross validation. Are the other experimental results not the cross validation results?
The final result reported in Table 3 are different from the results of Table 2. Are they separate ones?
If they are separate ones, I wonder how the author choose the optimal weight in experiments of Table 2.

Validity of the findings

The research presented very promising segmentation results.
However, it should be also noted that the number of the training case is not large enough.
So, I think this should be clearly reported in the manuscript.

Reviewer 2 ·

Basic reporting

- In summary, this paper presents a computational strategy for automatic segmentation of Epicardial fat (ECF) volume using deep learning computer vision, where the method employs non-contrast and contrast-enhanced CT images. It is a well-organized and well-written paper.

- Suggestion and Comments:

1- I would encourage the authors to clearly list the "Clinical Significance" and "Technical significance" of their work at somewhere within the "Introduction".

2- 3D UNet and even 3D attention-based UNet have been around for several years in the community. What kind of novelties this papers is carrying?

3- While for 2D images the UNet network(s) are computationally efficient, we know that 3D convolutions have huge storage requirements and therefore, end-to-end training is limited by GPU memory and data size. Have authors tried 2.5 UNet? If not, why?

4- The current work reports that it employed 15 radiologist to manually segment the data. However, I was not able to find the analysis of inter-rater agreement! I mean, the level of agreement among annotators. I would very much like to see the IoU and/or Dice coefficient to better understand the level of agreement.

5- How the authors did split the 220 patients to build a training, testing, and validation set?

6- I would encourage the authors to provide another comparative analysis using different number of slices plus different image scaling.

Experimental design

- Inter-rater agreement among those 15 radiologist is missing.

- Data stratification is missing.

- In general, the amount of data seems to be few, particularly to tackle one of the most difficult medical image segmentation tasks. Have they tried any type of data augmentation, at least for the training set?

- Is the improvement level statistically significance?

Validity of the findings

To me, this problem is one the most challenging problem in medical image analysis and interpretation. To me, I expect to see more in deep implementation, more data that covers more diversity, and a generalize model that can be applied across different healthcare centers.

---

## Round 0.2 · Minor Revisions

The reviewers are happy with your changes. I have read your submission carefully now and attach an annotated PDF with small suggested changes to the wording, etc. Please address these as much as possible and resubmit. If there is any specific question or comment that is unclear, feel free to email me at [email protected]. Please open the document in Acrobat Reader. Other PDF software may not show everything as intended.

Reviewer 1 ·

Basic reporting

The authors sincerely responded the reviewers' comments and modified the manuscript.
Thank you for the opportunity to review this manuscript.

Experimental design

I wish the authors could include the analysis of inter-rater agreement.
Except that, I have no comment.

Validity of the findings

no comment

Reviewer 2 ·

Basic reporting

The authors revised their manuscript accordingly based on my comments and recommendations.

Experimental design

The authors revised their manuscript accordingly based on my comments and recommendations.

Validity of the findings

The authors revised their manuscript accordingly based on my comments and recommendations.

---

## Round 0.3 · accepted · Accept

Thank you again for your submission and for even including inter-rater agreement in the latest version!

A couple of typos I noticed: "at of 300thh", "Weed test"